# Limited Association between Schizophrenia Genetic Risk Factors and Transcriptomic Features

**DOI:** 10.3390/genes12071062

**Published:** 2021-07-12

**Authors:** Alice W. Yu, J. David Peery, Hyejung Won

**Affiliations:** 1Department of Biostatistics, University of North Carolina at Chapel Hill, Chapel Hill, NC 27599, USA; alicewyu@live.unc.edu; 2Department of Computer Science, University of North Carolina at Chapel Hill, Chapel Hill, NC 27599, USA; jdavidp@live.unc.edu; 3Department of Statistics and Operations Research, University of North Carolina at Chapel Hill, Chapel Hill, NC 27599, USA; 4Department of Genetics, University of North Carolina at Chapel Hill, Chapel Hill, NC 27599, USA; 5Neuroscience Center, University of North Carolina at Chapel Hill, Chapel Hill, NC 27599, USA

**Keywords:** schizophrenia, GWAS, LD score regression, differentially expressed genes, co-expression networks, transcriptional regulation

## Abstract

Schizophrenia is a polygenic disorder with many genomic regions contributing to schizophrenia risk. The majority of genetic variants associated with schizophrenia lie in the non-coding genome and are thought to contribute to transcriptional regulation. Extensive transcriptomic dysregulation has been detected from postmortem brain samples of schizophrenia-affected individuals. However, the relationship between schizophrenia genetic risk factors and transcriptomic features has yet to be explored. Herein, we examined whether varying gene expression features, including differentially expressed genes (DEGs), co-expression networks, and central hubness of genes, contribute to the heritability of schizophrenia. We leveraged quantitative trait loci and chromatin interaction profiles to identify schizophrenia risk variants assigned to the genes that represent different transcriptomic features. We then performed stratified linkage disequilibrium score regression analysis on these variants to estimate schizophrenia heritability enrichment for different gene expression features. Notably, DEGs and co-expression networks showed nominal heritability enrichment. This nominal association can be partly explained by cellular heterogeneity, as DEGs were associated with the genetic risk of schizophrenia in a cell type-specific manner. Moreover, DEGs were enriched for target genes of schizophrenia-associated transcription factors, suggesting that the transcriptomic signatures of schizophrenia are the result of transcriptional regulatory cascades elicited by genetic risk factors.

## 1. Introduction

Genomic regions associated with schizophrenic risk have been identified through genome-wide association studies (GWASs) [1,2]. Hundreds of genetic variants associated with schizophrenia have been detected, making schizophrenia a highly complex and polygenic disorder [3]. A large proportion of these variants are located in the non-coding genome, and are thought to have a role in transcriptional regulation [1,4,5,6]. Moreover, schizophrenia risk genes include several transcriptional regulators [7,8,9,10], suggesting potential transcriptional dysregulation in schizophrenia. In line with these findings, there has been a series of studies measuring the expression signatures of postmortem brain tissue from schizophrenia-affected individuals. Transcriptional dysregulation has been observed, exemplified by a large number of differentially expressed genes (DEGs), differentially expressed transcripts (DETs), and co-expression networks associated with schizophrenia [11,12]. It has been hypothesized that schizophrenia-associated transcriptomic dysregulation is causally implicated with schizophrenia, and therefore directly influenced by genetic risk factors. While the association between schizophrenia-associated genetic risk factors and transcriptomic signatures has been consistently reported [8,11,13], the effect size of the observed association was small, suggesting that the contribution of genetic risk factors to transcriptomic dysregulation may be nominal. These results warrant systematic investigation of the relationship between schizophrenia genetic risk factors and transcriptomic features.

To address this, we assessed the contribution of different transcriptomic features to the heritability of schizophrenia. We first compiled sets of genes that are associated with different transcriptomic features in postmortem schizophrenic brain tissue, including differential expression, differential transcript usage, gene- and isoform-level co-expression networks, and network centrality. We then employed brain-derived functional genomic data, including chromatin interaction profiles (Hi-C) and gene regulatory variants (quantitative trait loci, QTLs) to identify the risk variants mapped to individual transcriptomic features [13]. We finally leveraged stratified linkage disequilibrium score regression (S-LDSC) on those risk variants to assess the contributions of different transcriptomic features to the heritability of schizophrenia. We hypothesized that discovering transcriptomic features that potentially contribute to the heritability of schizophrenia would help narrow down the expression measures that are directly impacted by the genetic risk factors.

We found that DEGs, DETs, and gene- and isoform-level co-expression networks show nominal association with genetic risk of schizophrenia. Notably, cell type-specific DEGs of three excitatory (Ex-L4, Ex-L4/5, and Ex-L5/6CCb) and two inhibitory (In-Rosehip and In-PV) neuronal subtypes showed strong association, suggesting that cellular heterogeneity may obscure the relationship between transcriptomic alterations and genetic risk factors. Moreover, genes with low network connectivity showed stronger heritability enrichment than genes with high network connectivity. These results collectively suggest that transcriptomic features may be indirectly impacted by the genetic risk of schizophrenia. In fact, DEGs were enriched for target genes of schizophrenia-associated transcription factors (TFs), demonstrating that the transcriptomic signatures of postmortem brain tissue with schizophrenia are the result of TF-mediated gene regulatory cascades.

## 2. Materials and Methods

### 2.1. Data Collection

Different features of the transcriptomic alterations observed in the postmortem brains of schizophrenic individuals were compiled from two sources—the PsychENCODE consortium (PEC) [11] and CommonMind consortium (CMC) [12]. These categories include differentially expressed genes (DEGs; PEC), differentially expressed transcripts (DETs; PEC), gene co-expression modules (PEC and CMC), isoform-level co-expression modules (PEC), and the two measures of central hubness of genes within a module: kME, which captures the centrality of a gene in a given module (PEC), and kTotal, which captures the overall connectivity between pairs of genes (CMC). Because PEC is the largest known dataset to date that encapsulates numerous transcriptomic datasets from schizophrenic individuals, we used PEC as the main data source. In addition to PEC, we also used CMC for co-expression modules to verify the results, because network membership and connectivity were defined differently between these two datasets.

DEGs and DETs were defined based on a false discovery rate (FDR) of less than 0.05 and further stratified into down- and upregulated genes/transcripts on the basis of the log2 fold change (logFC). If the logFC was positive, genes/transcripts were categorized as upregulated, and if negative, genes/transcripts were categorized as downregulated. For gene- and isoform-level co-expression modules, genes/transcripts were categorized based on the modules that represent a set of co-expressed genes. For kME and kTotal, we classified genes into ten groups based on the centrality and connectivity values, respectively, so that the genes within a certain percentile range of the kME/kTotal value were grouped together. Genes with the highest kME/kTotal values (hub genes) were assigned to the lowest percentile scores (e.g., 0–10%), and genes with the lowest kME/kTotal values (peripheral genes) were assigned to the highest percentile scores (e.g., 90–100%).

In addition to transcriptomic features obtained from bulk-brain RNA-seq, cell type-specific DEGs were obtained from Ruzicka et al. [14]. For a given cell type, we combined up- and downregulated genes to define DEGs, because there was a much smaller number of down- and upregulated genes identified by cell type-specific RNA-seq than were identified by bulk-brain RNA-seq. Therefore, to increase the statistical power of heritability enrichment analysis, we merged the down- and upregulated genes identified from each cell type to define cell type-specific DEGs.

From these data sources, we compiled a list of genes or isoforms for each transcriptomic feature. To identify the SNPs mapped to individual gene lists, we used three types of functional genomic data generated from the dorsolateral prefrontal cortex (DLPFC) by the PEC: Chromatin interaction profiles (Hi-C), expression quantitative trait loci (eQTLs), and splicing isoform quantitative trait loci (isoQTLs) [13]. Hi-C identifies genes with which the SNPs interact and QTLs identify SNPs that are correlated with gene or isoform expression levels.

### 2.2. S-LDSC Annotation File Generation

To generate the annotation files required for S-LDSC, each gene set was converted into an SNP list. We used the SNP–gene relationship of the H-MAGMA input file generated from the adult DLPFC [8] to identify the SNPs mapped to each gene set based on chromatin interactions. Because ~30% of eQTLs and isoQTL associations are supported by Hi-C evidence [13], we used the same H-MAGMA input file to map both genes and transcripts to SNPs.

The PEC eQTL resource provides SNPs that are associated with gene expression. We converted the genes to SNPs by identifying SNPs that show an association (FDR < 0.05) with a given gene. DEGs, gene co-expression modules, kME/kTotal groups, and cell type-specific DEGs were mapped to the SNPs on the basis of eQTL associations. Similarly, the PEC isoQTL resource provides SNPs that are associated with isoforms. Therefore, the DETs and isoform-level co-expression modules were converted to SNPs based on isoQTL association (FDR < 0.05).

SNPs were mapped to multiple genes when there were complex enhancer–gene interactions. On average, we found that 77.19 SNPs were annotated to a given gene, while 1.90 genes were linked to a given SNP in eQTLs. For Hi-C, an average of 73.57 SNPs were annotated to a given gene, while 1.02 genes were linked to a given SNP.

We then generated LDSC annotation files by overlapping our SNP lists with baseline LD annotation files (v1.1.0) [15]. These annotation files were deposited in the Won lab github repository (https://github.com/thewonlab/DEG_LDSC, (deposited on 11 July 2021)).

### 2.3. S-LDSC

LDSC is a computational framework that extracts multiple aspects of genetic architecture from GWAS. S-LDSC can be used to measure the heritability explained by a given genomic feature by partitioning SNPs into the regions of interest accounting for linkage disequilibrium (LD). Therefore, we performed S-LDSC (v1.0.1) using the generated LDSC annotation files [15]. The baseline model (v1.1.0) was used to assess heritability enrichment compared with the basic annotation of the genome. The S-LDSC results in heritability enrichment values and *p*-values that mark the significance of enrichment. Since genes have been previously shown to be enriched for heritability [15], high enrichment values are expected with the gene-centric approach that we undertook (e.g., DEGs). Thus, we mainly focused on the significance of enrichment by measuring the statistical significance of heritability enrichment (LDSC *p*-values) to confirm that the heritability enrichment explained by a given feature was not due to chance. When multiple gene sets were evaluated (e.g., kME/kTotal groups), we calculated FDR values using Benjamini and Hochberg (BH) to control for multiple tests. Gene sets with an enrichment value > 1 and an FDR < 0.05 were highlighted as enriched for schizophrenia heritability.

### 2.4. TF–Target Gene Linkage Analysis

We defined schizophrenia-associated TFs by identifying TFs out of high-confidence schizophrenia risk genes [13]. We then identified the target genes of schizophrenia-associated TFs based on TF–target gene linkages generated by the PEC [13]. Since PEC TF–target gene linkages only contain protein-coding genes, we ran Fisher’s exact test with protein-coding DEGs from the PEC and target genes of schizophrenia-associated TFs using all protein-coding genes as a background gene set. The DEGs were then stratified into up- and downregulated genes based on logFC, and the same analysis was repeated.

## 3. Results

### 3.1. Differentially Expressed Genes and Transcripts

Quantitative transcriptomic analysis on the postmortem brain tissue from individuals with schizophrenia has identified hundreds to thousands of genes that are differentially expressed in schizophrenia compared to neurotypical controls [11,12]. Thus, we first analyzed the contribution of DEGs to schizophrenia heritability.

We obtained DEGs from Gandal et al. [11] and partitioned them into sets of up- and downregulated genes. These gene sets were subsequently converted to a list of SNPs on the basis of SNP–gene relationships defined by eQTL and Hi-C datasets of the dorsolateral prefrontal cortex (DLPFC) [13]. S-LDSC analysis was performed on the DEG-assigned SNPs to quantify heritability enrichment driven by DEGs. We found that the SNPs assigned to downregulated genes via chromatin interaction profiles (Hi-C) showed heritability enrichment for schizophrenia (enrichment ± SE = 1.80 ± 0.27, *p* = $3.84\times 10^{-3}$, FDR = 0.015). On the contrary, SNPs assigned to upregulated genes did not display significant heritability enrichment (enrichment ± SE = 1.60 ± 0.33, *p* = $7.2\times 10^{-2}$, FDR = 0.11), suggesting that genetic risk factors for schizophrenia may contribute to the downregulation of genes in the postmortem brain samples of schizophrenia (Figure 1A). Surprisingly, SNPs assigned to DEGs via eQTL associations were not enriched for schizophrenia heritability in either up- or downregulated genes (Figure 1C).

Another distinct transcriptomic feature detected in schizophrenic brain tissue is the isoform-level dysregulation, which has been shown to be far more complex than gene-level dysregulation [11]. Therefore, we analyzed whether DETs can explain a significant portion of schizophrenia heritability. We exploited isoQTLs from the DLPFC [13] to identify those SNPs associated with differentially regulated isoforms. This list of SNPs was then used to perform S-LDSC analysis to quantify heritability enrichment explained by DETs. Notably, isoQTLs mapped to downregulated transcripts, but not upregulated transcripts, were nominally enriched for schizophrenia heritability (downregulated DETs, enrichment ± SE = 2.05 ± 0.52, *p* = $4.1\times 10^{-2}$, FDR = 0.083; upregulated DETs, enrichment ± SE = 1.83 ± 0.63, *p* = $1.9\times 10^{-1}$, FDR = 0.25) (Figure 1D). This is in stark contrast to the DEG-associated eQTLs that were not enriched for schizophrenia heritability (Figure 1C), suggesting that schizophrenia genetic risk factors may affect isoform usage. To further confirm this finding, we also acquired SNPs that physically interact with promoters and exons of DETs on the basis of Hi-C data [13]. Both down- and upregulated transcripts were enriched for schizophrenia heritability, but similar to isoQTL-based DET association, downregulated transcripts were more robustly enriched for schizophrenia heritability (downregulated DETs, enrichment ± SE = 1.67 ± 0.19, *p* = $3.48\times 10^{-4}$, FDR = 0.0028; upregulated DETs, enrichment ± SE = 1.58 ± 0.23, *p* = $1.21\times 10^{-2}$, FDR = 0.032) (Figure 1B). However, the extent of DET association with genetic risk of schizophrenia was modest. Collectively, these results demonstrate significant but weak relationships between schizophrenia genetic risk factors and differential expression signatures.

### 3.2. Co-Expression Networks

Differential expression signatures represent mixed biological processes that are abrogated in schizophrenia. We reasoned that co-expression networks could elucidate biological properties that are not easily detectable from DEGs. For example, co-expression networks delineate cell type-specific expression signatures (e.g., neuronal vs. glial modules) that are dysregulated in schizophrenia [11]. Furthermore, some schizophrenia-associated co-expression networks are associated with treatment response [16,17].

We therefore surveyed the contribution of schizophrenia-associated genetic risk factors to co-expression networks built from two consortia-level efforts—the CommonMind consortium (CMC [12]) and the PsychENCODE consortium (PEC [11]). CMC and PEC identified 4 and 20 co-expression modules with schizophrenia associations, respectively. As such, we analyzed the association between the CMC and PEC modules and heritability of schizophrenia.

Two PEC modules (geneM1 and geneM23) and two CMC modules (M1c and M2c) were significantly enriched for schizophrenia heritability when the genes were assigned to SNPs based on Hi-C evidence (FDR < 0.05; Figure 2). Among them, only M2c (enrichment ± SE = 2.63 ± 0.38, *p* = $3.48\times 10^{-5}$, FDR = 0.0012), a module implicated with neuronal and synaptic function, was enriched for genes differentially regulated in schizophrenia (odds ratio = 2.32, *p* = $1.04\times 10^{-13}$). This is consistent with previous findings that M2c is associated with the genetic risk factors of schizophrenia [12]. Moreover, geneM23 (enrichment ± SE = 5.23 ± 1.33, *p* = $1.6\times 10^{-3}$, FDR = 0.027), an interneuronal module, has been shown to be nominally downregulated in schizophrenia (β = −0.0024, *p* = $3.0\times 10^{-2}$), suggesting that genetic risk factors may contribute to the interneuronal dysfunction observed in schizophrenic brains [18]. It is also of note that another CMC module, M9c, was nominally enriched for schizophrenia heritability (enrichment ± SE = 6.77 ± 2.08, *p* = $5.8\times 10^{-3}$, FDR = 0.050). M9c has been shown to not only be enriched for schizophrenia DEGs (odds ratio = 3.94, *p* = $5.44\times 10^{-17}$), but also associated with copy number variation (CNV) in schizophrenia [12]. The biological processes represented by this module include glutamatergic synapse and mitochondrial functions [12].

Next, we evaluated schizophrenia heritability explained by SNPs mapped to each module based on eQTL associations. None of the PEC or CMC modules were enriched for schizophrenia heritability (Appendix A). Collectively, we found a moderate association between gene co-expression modules and schizophrenia genetic risk.

### 3.3. Isoform-Level Co-Expression Modules

In addition to gene-level co-expression networks, a complex picture of isoform-level co-expression networks has been identified [11]. Isoform-level co-expression networks showed greater disease association and specificity than gene-level co-expression networks, highlighting the need to investigate the relationship between schizophrenia heritability and isoform-level co-regulation.

Therefore, we analyzed 56 isoform-level co-expression modules defined by PEC via mapping transcripts to the corresponding SNPs either based on Hi-C or isoQTL evidence. When SNPs were assigned to isoforms using isoQTL associations, none of the isoform-level modules were enriched for schizophrenia heritability (Appendix A). On the contrary, we found one module (isoM2) to be enriched for genetic risk for schizophrenia when Hi-C mapping approach was used (Figure 3; enrichment ± SE = 2.37 ± 0.40, *p* = $7.3\times 10^{-4}$, FDR = 0.041). However, this module was not significantly dysregulated in the postmortem schizophrenic brain samples. Four additional isoform-level modules (isoM3, isoM14, isoM17, and isoM46) showed nominal enrichment for schizophrenia heritability (*p* < 0.05). Among these five isoform-level modules with genetic evidence, isoM14 and isoM17 were significantly downregulated in schizophrenia. Both isoM14 and isoM17 were neuronal modules, with isoM14 and isoM17 involved in GTPase and RBFOX1 signaling, respectively. Given the well-established role of RBFOX1 in neuronal alternative splicing programs [19], this result suggests that genetic risk factors for schizophrenia may regulate the master regulators of alternative splicing that leads to dysregulation of isoform networks.

### 3.4. Network Centrality

Since the contribution of genetic risk factors to schizophrenia-associated co-expression networks was nominal, we next evaluated other network properties that could be influenced by genetic risk factors. For example, module membership (kME) and network connectivity (kTotal) provide important metrics of network centrality. Genes with high centrality are called network hub genes and are thought to play a central role in a given network. Genes with low network centrality are called peripheral genes, and their expression is often influenced by hub genes. Network connectivities have been found to be associated with schizophrenia heritability, but only to the extent that the baseline LD model can predict [20].

This prompted us to analyze whether schizophrenia-associated genetic risk factors preferentially affect genes with high network centrality. To this end, we grouped genes with high (0–10%) to low (90–100%) kME [11] or kTotal [12] values regardless of their module membership. We then performed stratified LDSC using the SNPs mapped to the genes in each kME/kTotal group on the basis of Hi-C or eQTL evidence (Methods).

To our surprise, we found that genes with low network centrality were more likely to be affected by the genetic risk of schizophrenia when the Hi-C-based SNP–gene relationship was used (Figure 4). Although eQTL-based mapping did not yield any association between network centrality and schizophrenia heritability, we observed a similar pattern: Genes with low kTotal values are more highly enriched for schizophrenia heritability than genes with high kTotal values (Appendix A). This pattern from eQTL-based mapping corroborates what we observed from Hi-C-based SNP–gene mapping.

### 3.5. Cell Type-Specific Transcriptomic Signature

We reasoned that cellular heterogeneity may dilute the impact of genetic risk factors on transcriptomic signatures and account for the nominal relationship between schizophrenia heritability and global transcriptomic alterations. Therefore, we leveraged single-cell RNA sequencing (scRNA-seq) data from schizophrenic postmortem brain samples to assess whether the genetic risk factors of schizophrenia preferentially affect specific cell types [14]. We obtained cell type-specific DEGs and converted them to SNPs using SNP–gene relationships defined by eQTL associations or Hi-C interactions [13]. S-LDSC analysis was performed on the SNP lists to quantify the heritability enrichment driven by cell type-specific DEGs.

Similar to global transcriptomic analysis, cell type-specific DEGs did not show significant heritability enrichment when DEGs were assigned to SNPs via eQTL association (Appendix A). On the contrary, DEGs in a subset of excitatory (Ex-L4, Ex-L4/5, and Ex-L5/6CCb) and inhibitory (In-Rosehip and In-PV) neuronal subtypes showed significant heritability enrichment for schizophrenia via Hi-C interactions (Figure 5). Cortico-cortical projection neurons in layers 5–6 (Ex-L5/6CCb) showed the largest number of downregulated genes among excitatory neurons, and parvalbumin-expressing inhibitory neurons (In-PV) showed the largest number of dysregulated genes among inhibitory neurons, indicating that gene dysregulation in these cell types may arise from genetic risk factors.

These results align with the aforementioned findings. Ex-L4, Ex-L4/5, Ex-L5/6CCb, In-Rosehip, and In-PV display excess of downregulated genes in schizophrenic postmortem brain samples [19], which is in line with our finding that downregulated genes are more likely associated with schizophrenia heritability (Figure 1A). Moreover, gene co-expression modules enriched for schizophrenia heritability were implicated for neuronal and synaptic function (Figure 2), which corroborates our findings that a subset of excitatory and inhibitory neurons exhibit schizophrenia heritability enrichment. Collectively, these results suggest that cellular heterogeneity may obscure the detection of a reliable association between genetic risk factors and transcriptomic alterations.

### 3.6. Genetic Risk Factors Drive Transcriptomic Alterations through the Regulation of Transcription Factors

Given the limited association between schizophrenia heritability and transcriptomic alterations, we next investigated the possibility that transcriptomic signatures lie down-stream of the regulatory cascades elicited by the genetic risk factors of schizophrenia. For example, schizophrenia genetic risk factors may directly affect TFs, which in turn elicit transcriptional regulatory cascades that result in differential expression or a co-expression signature.

The role of TFs in schizophrenia has emerged. Transcriptional and epigenetic regulators show robust associations with schizophrenia GWASs [8,9,10]. Transcription factor 4 (*TCF4*), a master transcriptional regulator of schizophrenia risk genes and the genes involved in neural development and activity, is one of the schizophrenia susceptibility genes identified by GWASs [21,22,23]. Moreover, *SETD1A*, histone methyltransferase, is one of the high-confidence schizophrenia risk genes that harbor excess of rare loss-of-function variation [7]. These studies suggest that schizophrenia risk variants may affect TFs that have a broad impact on downstream transcriptomic architecture.

Thus, we interrogated the potential role of TFs in DEG. We first identified TFs among previously identified schizophrenia risk genes [13] to obtain those TFs that are potentially impacted by schizophrenia genetic risk (hereafter referred to as schizophrenia-associated TFs; Methods). We then leveraged TF–target gene regulatory networks built by PEC [13] to identify the target genes of schizophrenia-associated TFs. These target genes were then overlapped with schizophrenia DEGs to examine the possibility that DEGs are regulated by schizophrenia-associated TFs. Notably, DEGs from PEC were enriched for the target genes of schizophrenia-associated TFs (95% CI = 1.181–1.427, OR = 1.298, *p* = $7.342\times 10^{-8}$). When DEGs were stratified into up- and downregulated genes, both down- and upregulated genes were enriched for target genes of schizophrenia-associated TFs (downregulated, 95% CI = 1.039–1.343, OR = 1.183, *p* = $9.738\times 10^{-3}$; upregulated, 95% CI = 1.164–1.475, OR = 1.311, *p* = $8.4\times 10^{-6}$) (Figure 6). These results indicate that the transcriptomic alterations observed in postmortem schizophrenic brain tissue are likely the result of TF-mediated gene regulatory cascades.

## 4. Discussion

We analyzed different transcriptomic features to identify which features, if any, were impacted by the genetic risk factors of schizophrenia. The majority of transcriptomic features, including DEGs, DETs, and gene- and isoform-level co-expression modules, showed statistically significant heritability enrichment below a *p*-value threshold of 0.05. However, the evidence is weak that these transcriptomic features contribute to schizophrenia heritability, since many only barely surpassed this threshold without surviving corrections for multiple testing. Nominal heritability enrichment of DEGs in the brain homogenate could be partly explained by cellular heterogeneity, as the DEGs of neuronal subtypes, but not all cell types, were enriched for schizophrenia heritability. Among the excitatory neurons, the DEGs of (1) layer 4–5 excitatory neurons (Ex-L4 and Ex-L4/5) that form connections with the subcortical regions involved in sensory processing and (2) layer 5–6 corticocortical projection neurons involved in glutamatergic signaling (Ex-L5/6CCb) were enriched for heritability. It is of note that the dysregulated genes among Ex-L4, Ex-L4/5, and Ex-L5/6CCb showed high concordance [19], suggesting that dysregulation in excitatory neurons may have a common origin. Among the inhibitory neurons, we observed associations between the DEGs of Rosehip-expressing interneurons (In-Rosehip), the recently discovered inhibitory neuronal cell type that has not been previously discovered in mice [24], and schizophrenia heritability. This result corroborates our previous findings that human-specific gene regulation might be involved in neurodevelopmental and psychiatric disorders [25,26]. Parvalbumin-expressing interneuronal (In-PV) DEGs were also enriched for schizophrenia heritability, adding to the extensive evidence that In-PV are associated with schizophrenia [18,27,28,29].

In addition to differential expression signatures, we also observed an association between network connectivity and schizophrenia heritability. Surprisingly, we found that genes with low network centrality were enriched for heritability of schizophrenia. Given that we used the baseline LD model to predict schizophrenia heritability enrichment (Methods), this is consistent with the previous finding that the baseline LD model fully captures the heritability explained by network connectivity [20]. Furthermore, our findings support the recently proposed omnigenic model that cis-regulatory mechanisms likely perturb network peripheral genes more than core genes [30]. On the contrary, this enrichment could be due to the fact that peripheral genes are less likely to disrupt the system, so the common variants associated with them are more likely to be tolerated. Therefore, we cannot rule out the hypothesis that hub genes are still associated with more severe neurobiological phenotypes.

The weak association between schizophrenia heritability and transcriptomic features could also be explained by TF-mediated gene regulation. Indeed, the target genes of schizophrenia-associated TFs significantly overlapped with schizophrenia DEGs. This result indicates that the differential expression patterns observed in postmortem brains from schizophrenia-affected individuals are indirectly influenced by genetic risk factors through the regulatory cascades elicited by TFs.

Finally, our results pertain to schizophrenia, so whether this finding can be generalized to other psychiatric disorders warrants further investigation. In case nominal association between disease heritability and postmortem expression signatures is expanded to other disorders, it would be important to profile the transcriptional signatures across neurodevelopment in order to delineate the causal transcriptional programs impacted by the genetic risk factors of psychiatric illnesses [25]. Moreover, if the role of transcription factors on transcriptional signatures is reproduced in other disorders, trans-regulatory mechanisms will play an essential role in deciphering the genetic etiology of psychiatric illnesses.

## Figures and Tables

**Figure 1 genes-12-01062-f001:**
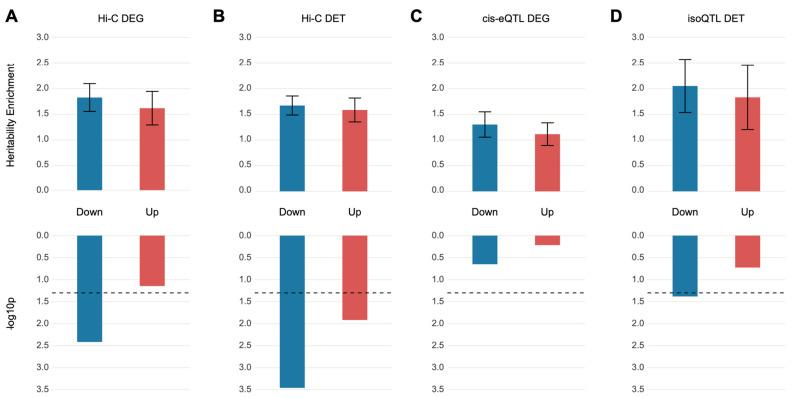
SNPs mapped to differentially expressed genes or transcripts are nominally enriched for heritability of schizophrenia. (**A**,**B**) Relationship between differentially expressed genes (DEGs) (**A**) and differentially expressed transcripts (DETs) (**B**) in schizophrenia postmortem brains and schizophrenia heritability when genes were mapped to SNPs on the basis of Hi-C evidence. Upregulated genes (enrichment ± SE = 1.60 ± 0.33, *p* = $7.15\times 10^{-2}$, FDR = 0.11) were not enriched for schizophrenia heritability, but the downregulated genes (enrichment ± SE = 1.80 ± 0.27, *p* = $3.84\times 10^{-3}$, FDR = 0.015) barely surpassed the FDR threshold (FDR < 0.05) and showed heritability enrichment of schizophrenia. Both downregulated (enrichment ± SE = 1.67 ± 0.19, *p* = $3.48\times 10^{-4}$, FDR = 0.0028) and upregulated (enrichment ± SE = 1.58 ± 0.23, *p* = $1.21\times 10^{-2}$, FDR = 0.032) transcripts showed significant enrichment for the heritability of schizophrenia. (**C**,**D**) Relationship between DEGs (**C**) and DETs (**D**) in schizophrenic postmortem brains and schizophrenia heritability when genes were mapped to SNPs on the basis of eQTL and isoQTL, respectively. Only downregulated DETs were nominally enriched for schizophrenia heritability (downregulated DETs, enrichment ± SE = 2.05 ± 0.52, *p* = $4.1\times 10^{-2}$, FDR = 0.083; upregulated DETs, enrichment ± SE = 1.83 ± 0.63, *p* = $1.9\times 10^{-1}$, FDR = 0.25).

**Figure 2 genes-12-01062-f002:**
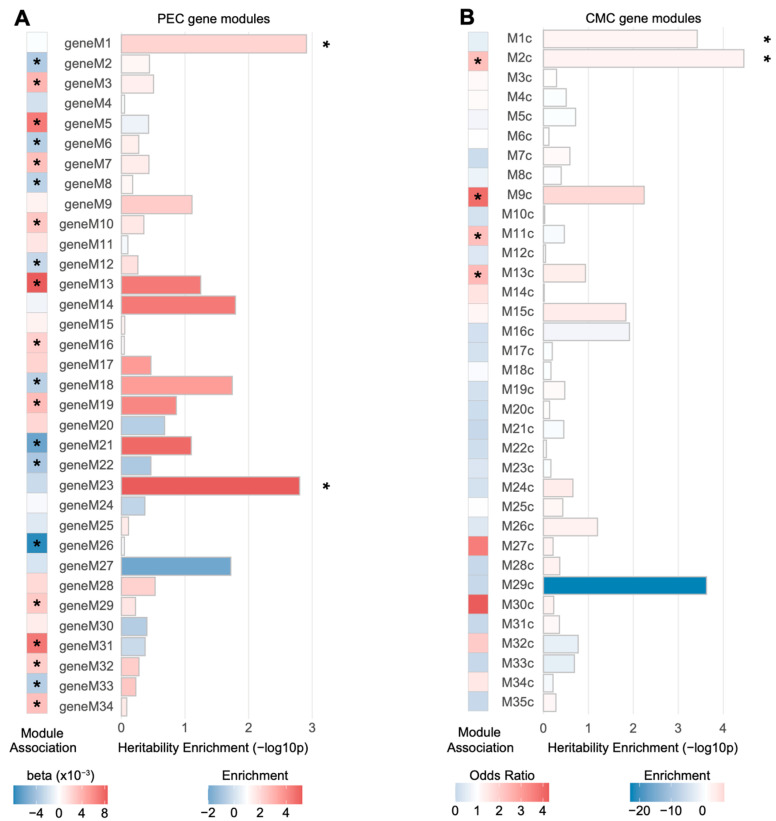
Schizophrenia heritability explained by gene co-expression modules associated with schizophrenia. (**A**) Two PEC gene-level co-expression modules, M1 and M23, were significantly enriched for schizophrenia heritability (* FDR < 0.05), while neither was significantly dysregulated in schizophrenia. Genes in each module were mapped to SNPs on the basis of Hi-C evidence and heritability explained by SNPs in each module was calculated. Co-expression modules marked with an asterisk (*) represent a PEC co-expression module significantly dysregulated with schizophrenia. Module association denotes whether the eigengene of a given module is upregulated (red, β>0) or downregulated (blue, β<0) in schizophrenia. (**B**) Two CMC gene-level co-expression modules, M1c and M2c, were significantly enriched for schizophrenia heritability (* FDR < 0.05). Among them, M2c was subject to dysregulation in schizophrenic postmortem brains. The genes in each module were mapped to SNPs on the basis of Hi-C evidence and the heritability explained by SNPs in each module was calculated. Co-expression modules marked with an asterisk (*) represent a CMC co-expression module enriched for DEGs in schizophrenia.

**Figure 3 genes-12-01062-f003:**
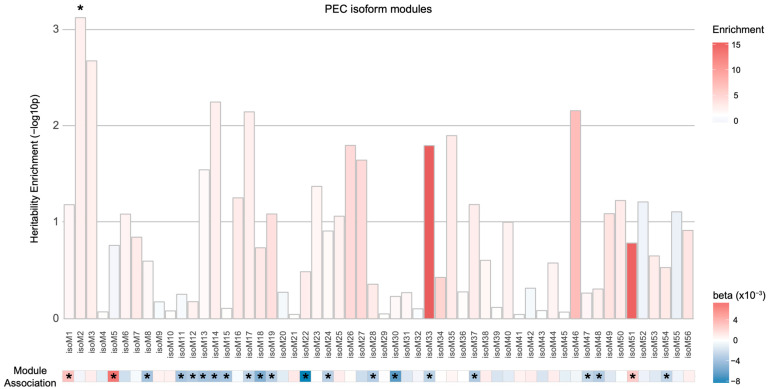
Heritability enrichment of the isoform-level co-expression modules associated with schizophrenia. One PEC isoform-level co-expression module (isoM2) was significantly enriched for schizophrenia heritability (* FDR < 0.05) when isoforms in the module were mapped to SNPs on the basis of Hi-C evidence. However, isoM2 was not significantly dysregulated in schizophrenia. An asterisk (*) in the module association heatmap indicates the isoform co-expression modules that are significantly associated with schizophrenia with eigengenes upregulated (red, β>0) or downregulated (blue, β<0) in schizophrenic postmortem brains.

**Figure 4 genes-12-01062-f004:**
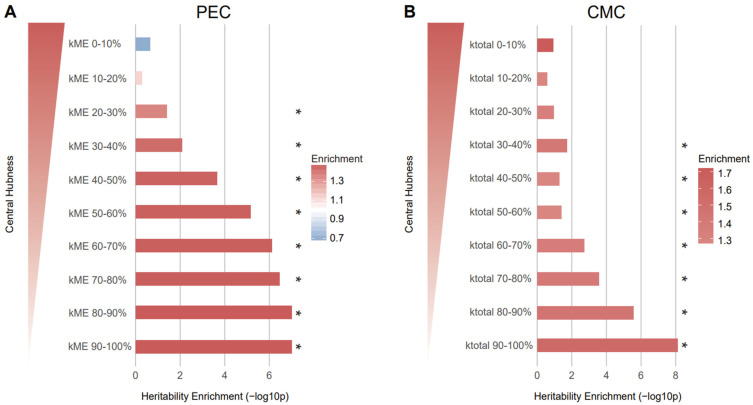
Genes with low network connectivity were enriched for schizophrenia heritability. (**A**) Genes in the kME 90–100% group (enrichment ± SE = 1.50 ± 0.09, *p* = $9.88\times 10^{-8}$, FDR = $4.95\times 10^{-7}$) and kME 80–90% group (enrichment ± SE = 1.50 ± 0.09, *p* = $9.9\times 10^{-8}$, FDR = $4.95\times 10^{-7}$) were most enriched for heritability of schizophrenia. (**B**) Genes in the kTotal 90–100% group (enrichment ± SE = 1.57 ± 0.1, *p* = $7.84\times 10^{-9}$, FDR = $7.84\times 10^{-8}$) were most enriched for heritability of schizophrenia, indicating that peripheral genes are more associated with heritability. The genes in each kME/kTotal group were mapped to SNPs on the basis of Hi-C evidence and the heritability explained by the SNPs in each group was calculated. As kME/kTotal increases, the central hubness of genes within a given module increases. * FDR < 0.05.

**Figure 5 genes-12-01062-f005:**
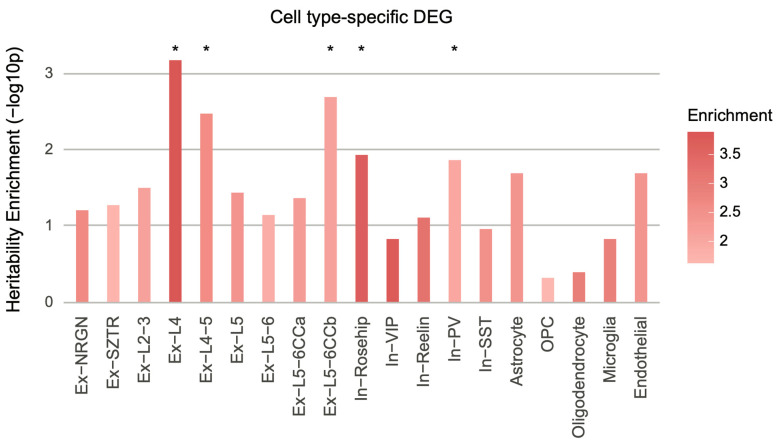
Cell type-specific schizophrenia heritability enrichment. Cell type-specific schizophrenia DEGs were mapped to SNPs on the basis of Hi-C interaction profiles, which were subsequently used to calculate schizophrenia heritability enrichment. DEGs in excitatory neurons (Ex-L4, enrichment ± SE = 3.89 ± 0.84, *p* = $6.28\times 10^{-4}$, FDR = 0.012; Ex-L4/5, enrichment ± SE = 2.64 ± 0.55, *p* = $3.2\times 10^{-3}$, FDR = 0.02; Ex-L5/6CCb, enrichment ± SE = 2.14 ± 0.37, *p* = $1.9\times 10^{-2}$, FDR = 0.018) and inhibitory neurons (In-Rosehip, enrichment ± SE = 3.73 ± 1.07, *p* = $1.1\times 10^{-2}$, FDR = 0.049; In-PV, enrichment ± SE = 2.02 ± 0.41, *p* = $1.3\times 10^{-2}$, FDR = 0.049) were significantly enriched for heritability of schizophrenia. Ex, excitatory neurons; In, inhibitory neurons; L, layer; NRGN, neurogranin; SZTR, schizophrenia transcriptional resilience; CC, cortico-cortical; VIP, vasoactive intestinal polypeptide; PV, parvalbumin; SST, somatostatin; OPC, oligodendrocyte progenitor cells. * FDR < 0.05.

**Figure 6 genes-12-01062-f006:**
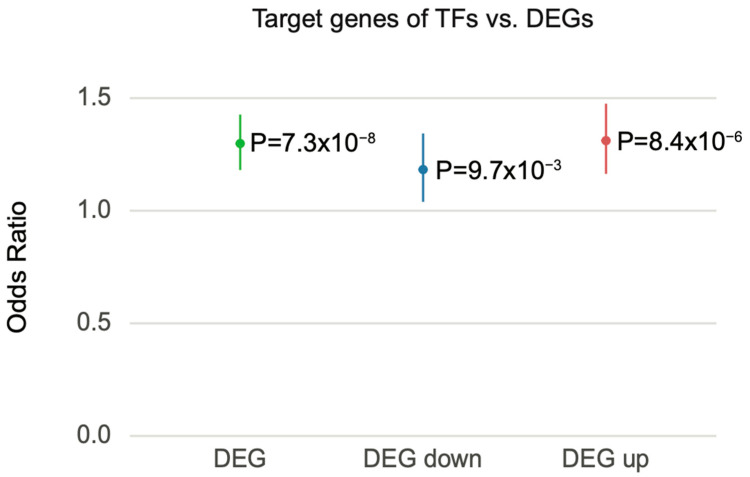
Schizophrenia DEGs were overrepresented with the target genes of schizophrenia-associated TFs. Forest plot of odds ratio estimates with 95% confidence intervals from overrepresentation analysis between schizophrenia DEGs and the target genes of schizophrenia-associated TFs. DEG represents a combined list of up- and downregulated genes.

## Data Availability

The LDSC annotation files are available through the Won lab github repository (https://github.com/thewonlab/DEG_LDSC, (deposited on 11 July 2021)). Hi-C-based SNP–gene relationships were obtained from https://github.com/thewonlab/H-MAGMA, (accessed on 09 March 2020) (Adult_brain.genes.annot). PEC eQTLs (DER-08a_hg19_eQTL.significant), isoQTLs (DER-10a_hg19_isoQTL.significant), DEGs (DER-13_Disorder_DEX_Genes), DETs (DER-14_Disorder_DEX_Transcripts), gene-level co-expression modules and module membership (DER-16_Disorder_Gene_Modules), isoform-level co-expression modules (DER-17_Disorder_Isoform_Modules), transcription factor–target gene linkages (INT-11_ElasticNet_Filtered_Cutoff_0.1_GRN_1), and schizophrenia risk genes (INT-17_SCZ_High_Confidence_Gene_List) were also obtained from http://resource.psychencode.org/, (accessed on 14 December 2018). CMC DEGs, gene-level co-expression modules, and module membership were obtained from Appendix A of Fromer et al. [12].

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
