# Peer review of "Limited Association between Schizophrenia Genetic Risk Factors and Transcriptomic Features"

_genes, 2021, doi:10.3390/genes12071062_

Round 1

Reviewer 1 Report

This manuscript by Yu et al aims to assess which gene or regulatory properties account for heritability of schizophrenia.  They conclude that differentially-expressed genes and co-expression networks show modest enrichment, with some cell-type specificity.  They instead attribute transcriptomic signatures to targets of schizophrenia-associated transcription factors.  It is a very dense paper with some interesting results but insufficient context to fully interpret them.

1) Throughout the results, the authors fail to distinguish effect size from significance, and to interpret negative results with respect to power.  For example, in the analysis of DEGs/DETs, much is made of the significance or not of upregulated vs. downregulated, but the enrichment of 1.6 vs. 1.8 or 1.8 vs. 2 appear quite similar.  The authors suggest these results are in line with postmortem results for gene downregulation in schizophrenia brains (line 335).  However, is there any relevant context to these observations, i.e. does the same proportion of genetic variants can cause downregulation and upregulation, in general?  For coding variation, it is well known that loss-of-function mutations are much more common than gain-of-function so if the pattern is similar for regulatory variation it may be relevant.  The overall interpretation is "weak relations between schizophrenia genetic risk factors and differential expression signatures".  However, the authors interpret the enrichment of targets of scz-associated TFs as meaningful, with ORs of 1.2-1.3.  It's difficult as a reader to follow the reasoning across the study or interpret 'enrichment', i.e. in comparison to variance explained.

2) The two main resources utilized, CMC and PEC, are not described, particularly in comparison to each other.  For example, some of the analyses are only carried out for one or the other, presumably due to limitations or advantages of one of the datasets but it is not clear to the reader.  Further, it would be helpful in interpreting results that appear to differ substantially, such as the co-expression networks (Fig 2) - why do the authors believe the profiles are not more similar?

3) The description of the annotation could use additional explanation, as it is critical to the results.  How is overlap handled, e.g. one SNP annotated to multiple genes, etc.?

4) Similarly, when the authors state that the targets of scz-associated TFs are enriched in DEGs, what is the comparison dataset?  What are the relevant variables to match - this may be well-served by a permutation mechanism to compare targets of scz-associated TFs to targets of non-associated TFs, other brain-expressed genes, etc.  Were these tested for heritability enrichment, as well?  If the authors are presenting it as an alternative explanation, they need to lay out a more complete picture of the limitations of the categories they have tested vs. possibilities.  Although the study was framed here as trying to 'explain heritability' of schizophrenia, perhaps the opposite framing of trying to 'explain' transcriptomic differences would be more consistent with the study design? 

5) In Figure 1, the graphs depict P-value but the description suggests FDR - it would be helpful to clarify where/why each approach is used.

6) The authors use the term 'response' to describe the targets of TFs that might be dysregulated by genetic variation impacting TFs.  The term I would use is 'downstream', as I would see this kind of impact as a consequence rather than a 'response' to the genetic perturbation of a TF.  (To me, 'response' indicates feedback or compensation mechanisms.)

7) Overall, the study could benefit from more contextualization of the results.  For example, without any comparison it's difficult to interpret the level of heritability enrichment observed.  Are there some gold-standard sets of loci that could frame this, such as 'synaptic' genes, brain-expressed genes, complement pathway, or other well-established categories that could serve as positive controls?  Likewise, it might be helpful to add some negative controls, such as DEGs in Alzheimer's or another brain disorder not thought to be etiologically similar to show that the enrichment observed is meaningful.  Finally, the finding that SCZ genes are peripheral to gene networks could use additional context or follow-up - what is the implication of this, and is there another testable prediction or comparison to be made?  For example, are there any diseases known to be impacted by central hub genes for comparison?  Does this imply limited gene dysregulation would be expected in schizophrenia?  More polygenicity?  How do known highly penetrant genetic causes of SCZ fit in?

Although this manuscript reports some intriguing findings and the approach of heritability enrichment is nice, the reader comes away with a list of observations but insufficient context to interpret them with respect to schizophrenia etiology.

Author Response

We appreciate the reviewer’s constructive feedback. Please see attached for our point by point response.

Reviewer 2 Report

The authors of this manuscript present a statistical approach to test if genetic heritability for SCZ is explained by genetic variation regulating transcription of genes and or isoforms or mapping to genes associates with altered expression in SCZ. Based on transcriptomic studies and sLDSC the authors show that the variants mapping to the respective differentially expressed gene, isoforms or gene networks were in parts enriched for heritability of schizophrenia and specifically for targets of Transcription Factors associated with SCZ

Overall the article is very well written and I only have very minor comments to make 

Due to the many abbreviations reading I would recommend adding an overview Figure of the Samples and analysis performed. This might make it easier to follow the different though converging approaches. 

The title in my opinion only captures one aspect of the full analyses and was a bit misleading for me, as I expected Transcription factors to be more at the core, whereas to my understanding genetic variants and eQTLs were the features the heritability estimates were based on. 

In the discussion, I do not fully agree with the assumption that peripheral genes have more impact, rather I would assume that peripheral genes are less likely to disrupt the system, in contrast to central hub genes, which are enriched for essential genes (https://doi.org/10.1371/journal.pone.0005344)
I thus tend to favour the hypothesis that in psychiatric disorders disruption of hub genes is associated with more severe neurobiological phenotypes. 

Author Response

We appreciate the reviewer’s positive and constructive feedback. Please see attached for our point by point response.
